



**1** **Is the content and potential preservation of soil organic**

**2** **carbon reflected by cation exchange capacity? A case study**

**3** **in Swiss forest soils**

Emily F. Solly[1], Valentino Weber[1], Stephan Zimmermann[2], Lorenz Walthert[2], Frank
Hagedorn[2], Michael W. I. Schmidt[1]
[1] University of Zurich, Department of Geography, Winterthurerstrasse 190, 8057 Zürich,
Switzerland.
[2] Swiss Federal Institute for Forest, Snow and Landscape Research WSL, Zürcherstrasse 111,
8903 Birmensdorf, Switzerland.
Correspondence to: E. Solly (emily.solly@geo.uzh.ch)

**14** **Abstract**

The content of organic carbon (C) in soils is not stable, but depends on a number of
environmental variables and biogeochemical processes that actively regulate its balance. An
improved identification of the environmental variables that can be used as predictors of soil
organic C (SOC) content is needed to reduce uncertainties of how the soil C reservoir will
respond to environmental change. Although several simulations rely on the amount of clay to
reproduce changes in the balance of SOC, recent efforts have suggested that other soil
physicochemical properties may serve as better predictors. Here we tested whether the effective
cation exchange capacity (CEC eff.), may be a more suitable predictor of the content and
potential preservation of SOC as compared to the mere quantification of clay-size particles. We
further assessed how various climatic, vegetation and edaphic variables explain the variance of
SOC content across different soil depths and soil pH classes. A set of more than 1000 forest





sites across Switzerland, spanning a unique gradient of mean annual precipitation (636-2484
mm), altitude (277-2207 m a.s.l), pH (2.8-8.1) and representing different geologies and soil
orders was used as a case study for this linear model analysis. Our results showed that CEC eff.
has the largest explanatory potential of SOC content (35 % of response variance in the complete
mineral soil profile) as compared to the amount of clay (which only explained 7 % of the
response variance in the complete mineral soil profile) and other environmental variables. CEC
eff. is strongly linked to SOC especially in the top mineral soil (0-30 cm depth) with the larger
presence of organic matter. At deeper soil depths most of the variance in SOC is instead
explained by climate, which in Switzerland is related to a greater weathering activity and
translocation of organic C through leaching with increasing mean annual precipitation. We
further observed soil pH to have a complex influence on SOC content, with CEC eff. being a
dominant variable controlling SOC content at pH >4.5 in the upper mineral soil and pH >6 in
the subsoil. Since CEC eff. is an edaphic property which is intimately associated to both the
conditions that shaped the soil and the current edaphic physicochemical conditions, these
findings indicate that considering CEC eff. as an integrative proxy for the potential preservation
of SOC and its alteration could improve future predictions of how the soil C reservoir will feed
back to environmental change.

## 1 Introduction

Large uncertainties in our understanding and predictions of how the terrestrial carbon (C) cycle
interacts with alterations in the environment exist about the effects on the content of soil organic
C (SOC) and its stabilization or destabilization (Jobbágy and Jackson, 2000; Friedlingstein et
al., 2014; Jackson et al., 2017; Todd-Brown et al., 2013). To assess the impacts not only of the
occurring but also of the forecasted global changes, it is desirable to identify environmental
predictors of SOC dynamics (Bailey et al., 2018; Rasmussen et al., 2018; Harden et al., 2018),
that match the recent shifts in paradigm of SOC (de)stabilization (Lehmann and Kleber, 2015;



Schmidt et al., 2011). Measuring and predicting the content and potential preservation of SOC
is elaborate due to the large spatial and temporal scales that are needed to detect changes
(Harden et al., 2018; van der Voort et al., 2016). However, the exchange of knowledge between
empirical research and simulation models offers great promise to facilitate innovations in C
management and programs to reach climate-change mitigation goals, e.g. the '4 per mille Soils
for Food Security and Climate' international research program (Minasny et al., 2017).

Advances in soil science have demonstrated that the content and preservation of SOC is

not controlled by the chemical composition of organic matter alone but is rather predominantly
driven by environmental and biological variables (Schmidt et al., 2011; Marschner et al., 2008;
Kleber and Johnson, 2010). In this context, the effects of climatic, biotic, and geogenic controls
of SOC dynamics have been extensively studied for different spatial and temporal scales
(Davidson and Janssens, 2006; Hicks Pries et al., 2017; Jobbágy and Jackson, 2000; Falloon et
al., 2011; Liang et al., 2017; Doetterl et al., 2015). Climatic variables generally appear to exert
a major control of SOC dynamics and consequently a significant feedback of the terrestrial-C
cycle to climate change is expected (Carvalhais et al., 2014; Chen et al., 2013). Biotic activity
(plant and microbial) and soil physicochemical variables are additionally contributing to
explain the large uncertainties covering the fate of SOC (Doetterl et al., 2015; Schmidt et al.,
2011; Rasmussen et al., 2018; Torn et al., 1997).

Despite this improved understanding, model frameworks aimed at assessing how SOC

responds to environmental change still necessitate the use of proxy variables that represent soil
characteristics that cannot be measured (correlative proxies), or that aggregate information
about multiple soil characteristics (integrative proxies)  (Bailey et al., 2018). To improve
projections of the largest actively cycling terrestrial C reservoir to environmental change, recent
efforts have hence been focusing on identifying new sets of variables that can be used to
determine the content of SOC. For instance, although several biogeochemical models rely on
the amount of clay to simulate mineral protection of SOC (Coleman and Jenkinson, 1996;



Wieder et al., 2015), Rasmussen et al. (2018) recently suggested that other soil physicochemical
properties, such as exchangeable Ca or Fe, and Al oxyhydroxides, can be used as better
predictors depending on the local soil pH. Soil pH reflects the chemical state of soil systems,
e.g. protonation, controlled by geological and mineralogical properties, and dictates some of
the main driving processes of SOC (de)stabilization (Deng and Dixon, 2002; Oades, 1988).
Such processes include organo-mineral complexation, solubility and organism activity, which
vary with depth across soil profiles (as reviewed by Sollins et al., 1996; Six et al., 2004;
Trumbore, 2009).

The amount of clay, that is the soil mineral fraction <2 μm in size, influences the content

of SOC by promoting the adsorption of organic molecules to its surfaces primarily by the
interaction with pedogenic oxides and the aggregation within clay structures (Lützow et al.,
2006; Oades, 1988; Eusterhues et al., 2003). Through its control on soil hydrology, oxygen
availability as well as soil microbial community, the amount of clay can further indirectly alter
SOC preservation (Andrews et al., 2011; Fierer and Schimel, 2002). Due to these properties,
the amount of clay is often used as a proxy for sorption on mineral surfaces and aggregation of
SOC, especially by more generalized larger scale models (Bailey et al., 2018). Several recent
studies, however, did not find the amount of clay to be a predominant physicochemical predictor
of SOC content and preservation (e.g. Schrumpf et al., 2013; Rasmussen et al., 2018; Doetterl
et al., 2015). A reason for this is likely that the amount of clay merely represents a size class of
soil particles rather an equivalent indication of available surface area or aggregate formation. It
follows that other soil properties that represent an indication of available soil surfaces may be
more appropriate to estimate SOC than clay.

An example of a soil property representing an indication of available soil surfaces is the

effective cation exchange capacity (CEC eff.). CEC eff. is an edaphic property reflecting
geology, mineralogical composition, organic matter and the pH that shapes the soil and its
current physicochemical conditions. The CEC eff. of a soil represents the total amount of




exchangeable cations such as $Na^+$, $K^+$, $Mg^{2+}$, $Ca^{2+}$, $Mn^{2+}$, $Al^{3+}$, $Fe^{2+}$, $H^+$ that can be retained
through electrostatic adsorption on soil particle surfaces. Soil particles exhibit negative and
positive charges that can adsorb oppositely charged ions from the soil solution. In most soils,
CEC eff. increases with soil pH. At low pH, it is mainly the permanent charges of the 2:1 type
clays that adsorb exchangeable cations, while at higher pH values the negative charges on some
1:1-type clays, allophane, Fe- and Al- oxides, and soil organic matter increase, thereby
increasing CEC eff. (Weil and Brady, 2016). There are several reasons why CEC eff. may be a
better integrative proxy for the content and potential preservation of SOC than others.
Quantifying the amount of clay is time-consuming. Determining clay mineralogy is expensive
and requires specialized equipment, and quantification can be challenging. The key variables
for soil organic matter protection are short range order minerals, Al- and Fe- oxyhydroxides,
and Al-, Fe- organo-metal complexes (Rasmussen et al., 2018), which cannot be thoroughly
identified and quantified for large sample sets with analytical techniques available today. Soil
surface area that has been calculated by using gas adsorption as a proxy for the protective
potential of soil and sediment, cannot be measured routinely on large sample sets. On the other
hand, CEC eff. is measured routinely to assess soil fertility for agricultural and forest use.
Here we ask the question whether CEC eff. can serve as a new integrative proxy of SOC
content and its potential preservation, integrating effects of soil surfaces of different clay types,
short-range order mineral phases and organic matter at the actual soil pH. Moreover, we are
interested in understanding how the variance of SOC content is explained by various
environmental variables across different soil depths and soil pH classes. We test this by using
a set of more than 1000 forest soil profiles across Switzerland spanning a strong gradient of
climate, altitude, geology, soil pH and soil orders (Gosheva et al., 2017).

**2 Materials and Methods**
**2.1 Study area and dataset**





The study area covers the complete country of Switzerland (~ 45-47° N, 6-10° E), situated in
the centre of Europe. Switzerland offers a wide and intricate range of geology and topography
which vary abruptly often within short distances, leading to very diverse soil orders (Walthert
et al., 2013; Gosheva et al., 2017). About 30 % of the country (~ 12.000 km$^2$) is covered by
forest, and half of this area is located above 1000 m a.s.l. Forest management is mainly practiced
at low elevations, where no large-scale clear-cutting is applied and natural regeneration is often
fostered by silvicultural management (Brassel and Brändli, 1999). Forest soil fertilization and
liming was always forbidden in Switzerland.
The dataset analyzed in this study originates from a database of the Swiss Federal
Institute for Forest, Snow and Landscape Research (WSL) containing data on 1204 forest sites
across Switzerland (for more details see Walthert et al., 2013; and Walthert and Meier, 2017).
More than 80% of these sites have been covered for at least 150 years with forests (Gosheva et
al., 2017).
The variables which we evaluated included SOC content (g kg$^{-1}$) and other soil
physicochemical properties (pH in CaCl$_2$, CEC eff. (mmolc kg$^{-1}$) and soil texture) as well as
climatic, topographic and vegetation data. In this study we report weighted means of SOC
content over fixed depths rather than SOC stocks. This is because SOC content represents a
direct measure of SOC after a single correction for the amount of fine earth in pedogenic
horizons, while SOC stock estimates required multiple imponderable corrections (Gosheva et
al., 2017).

**2.2 Soil chemical and physical properties**
In each forest site a soil profile was sampled by pedogenetic horizons down to a profile depth
of 120 cm if possible, otherwise down to parent rock, with six samples per pit on average. Soil
samples were dried at 40-60° C and sieved at 2 mm for chemical analyses. SOC content was
measured in milled subsamples by dry combustion using a C/N analyser (NC 2500, Carlo Erba



Instruments, Italy). Inorganic C was removed in samples with a pH above 6.0 by fumigating
with HCl vapour prior to analysis (Walthert et al., 2010). Soil pH was measured
potentiometrically in 0.01 M $CaCl_2$ with a soil - extract ratio of 1:2 after 30 min of equilibration.
Exchangeable cations were extracted in an unbuffered solution of 1 M $NH_4Cl$ for 1 h on an end-
over-end shaker using a soil - extract ratio of 1:10. The elemental concentrations were
subsequently measured with an ICP-AES (Optima 3000, Perkin–Elmer). For soil samples with
a pH ($CaCl_2$) <6.5 concentrations of exchangeable protons were calculated as the difference
between the total and the Al-induced exchangeable acidity with the KCl method (Thomas,
1982). For soil samples with a higher pH, concentrations of exchangeable protons were
assumed to be negligible. The effective cation exchange capacity (CEC eff.) was finally
calculated by summing the charge equivalents of exchangeable Na, K, Mg, Ca, Mn, Al, Fe and
H. Grain size distribution was measured with the sedimentation method according to Gee and
Bauder (1986) for 750 soil profiles. For the remaining soil profiles, we used the field estimates
based on ten texture classes from Walthert et al. (2004). We focused on the following soil-depth
intervals: 0-30 cm of the mineral soil profile (topsoil), 30-120 cm of the mineral soil profile
(subsoil) and for a comprehensive interpretation of the data we also considered the complete
mineral soil profile at 0-120 cm. We calculated the mean average content of all the above
mentioned soil properties for 0-30 cm, 30-120 cm and 0-120 cm by weighting the averages of
the content with the amount of fine earth determined in the pedogenetic horizons of those fixed
depth increments, as described in Walthert et al. (2013). Soil type was classified according to
the World Reference Base from 2007 (IWG, 2007).

### 2.3 Climatic, topographic and vegetation data

The soil data of each forest site was paired with climatic and vegetation data. Climatic data was
based on the Swiss metereological network (MeteoSwiss), combined with suitable interpolation
algorithms (Walthert et al., 2013). Specifically, mean annual precipitation (MAP) and mean





annual temperature (MAT) were provided by Meteotest (https://meteotest.ch/en/) for the period
1981-2010 (for details see Remund et al., 2014). The altitude of the forest sites was extracted
from a 25 m digital elevation model (DEM) of the Federal Geo-Information centre swisstopo.
For each forest site, the local Leaf Area Index (LAI) a common measure of canopy foliage
acting as a control over primary production (Asner et al., 2003), was further estimated according
to Schleppi et al. (2011) based on data from a vegetation survey using the Braun-Blanquet cover
abundance scale (Braun-Blanquet, 1964; Mueller-Dombois and Ellenberg, 1974).

**2.4 Statistical analysis**
To identify which environmental variable of interest had the greatest prediction power on SOC
content in the complete mineral soil profiles (0-120 cm), topsoils (0-30 cm) and subsoils (30-
120 cm), we applied linear model analysis. CEC eff., percentage of clay, LAI, MAT and, MAP
were treated as explanatory variables while SOC content was treated as main response variable.
To meet the normality assumptions of the applied statistical tests and standardize the variation
among variables, we transformed the continuous variables to normal distributions applying log-
transformations when required, and subsequently standardized them to a mean of 0 and standard
deviation of 1. The linear models were checked for multicollinearity using the variance inflation
factor (VIF). The value of VIF >4 was set as a threshold for evidence of multicollinearity. We
checked the model assumptions using the diagnostic plot functions in R (Crawley, 2012), and
the normality of the residuals was tested using histograms. The explanatory power of the
variables included in the linear models was computed as the standardized regression coefficient.
The more positive or negative the coefficient is, the higher is its relative power in predicting
the response variable SOC. We then partitioned the contribution of each of the explanatory
variables to the explained variance in SOC employing hierarchical variance partitioning. For
this, we used the 'lmg' metric in the R package 'relaimpo' (Groemping and Matthias, 2006),
which decomposes R squares into non-negative contributions that automatically sum to the total





R squared of the linear model and takes care of the dependence on orderings by averaging over
orderings (Grömping, 2006). The contribution of the environmental variables in explaining the
variance of SOC was additionally normalized to sum to 100 %, i.e. in place of the total R
squared (Groemping and Matthias, 2006). We further performed a linear model analysis of the
relative importance of climatic, vegetation and edaphic variables to predict SOC across pH
classes at 0-120 cm, 0-30 and 30-120 cm soil depth. Specifically, soil-pH data was grouped into
nine consecutive pH ranges: (pH: <4, 4-4.5, 4.5-5, 5-5.5, 5.5-6, 6-6.5, 6.5-7, 7-7.5, >7.5 see
Table 1 for the complete pH ranges at the three different depths). The relation between SOC
content and all explanatory variables, parsed among different soil depths and pH classes, was
also tested using Spearman's correlation coefficient. Significance was evaluated based on
Bonferroni's corrected p-values to account for Type I errors. For each statistical test $P < 0.05$
was considered to be statistically significant. All analyses were performed with the R statistical
software version 3.5.0  (R Core Team, 2018), and the packages 'vegan', 'MASS', 'dplyr',
'plyr', 'car', 'quantPsyc','caret','relaimpo', 'Psych'.

**3 Results**
**3.1 Environmental characteristics of the forest sites**
The Swiss forest sites where the soils were sampled are distributed between 277 and 2207 m
a.s.l. They are characterized by a mean annual precipitation (MAP) ranging between 636 and
2484 mm, and mean annual temperatures (MAT) ranging between 0.1 and 12.0 ° C. MAT and
altitude are strongly negatively correlated (r = -0.97). The predominant soil types are classified
as Cambisols (n = 365), Luvisols (n = 127) and Gleysols (n = 89). About half of the sites are
covered by broadleaf tree species while the rest are coniferous forests (Gosheva et al., 2017),
with LAI values varying between 2 to 7.2. pH values range from 2.8 to 8.1 and the percentage
of clay varies between 0.8 and 75.6 % (for details about specific ranges in topsoils and in the
subsoils see Table 1). The parent material of the north of Switzerland is dominated by a





calcareous bedrock and sediments, from which alkaline soils have developed. The most acidic
soils are instead found in the Southern Alps and parts of the Swiss plateau, in the center of
Switzerland (Figure S2). The content of SOC varies between 6 and 376.8 g kg$^{-1}$ in topsoils
(upper 0-30 cm of the mineral soil) and between 0 (lower than detection limit) and 229 g kg$^{-1}$
in subsoils (30-120 cm depth). CEC eff. ranges between 20.4 and 1046.4 mmolc kg$^{-1}$ in topsoils
and between 6.2 and 727.9 mmolc kg$^{-1}$ in subsoils.

### 241    3.2 Predictors of soil organic carbon (complete pH range)

All of the explanatory variables except MAT in the topsoil have a significant effect on SOC
content when considering the complete pH range (Table 2). The standardized regression
coefficients indicate that throughout Switzerland the SOC content increases with increasing
CEC eff. and MAP, while decreases with greater MAT, LAI and percentage of clay. The signs
of these relations are overall coherent with the sign of significant Pearson's correlations
coefficients, with the exception of the correlation between SOC and CEC eff. in the pH range
4.5-5 in the subsoil (Fig. 1). The assessment of the relative contribution of the explanatory
variables to the explained variance in SOC shows that within the complete mineral soil profile
CEC eff. explains most of the response variance in SOC content (35 %), followed by MAP (26
%), and smaller significant contributions of explained variance by LAI and MAT (Table 2).
Within the topsoil CEC eff. also has the overall strongest influence on SOC accounting for 60
% of its variance (Table 2). In subsoils, climatic variables (MAP and MAT) are instead the
predominant variables controlling the variance in SOC content (31 and 28 % respectively)
(Table 2).

### 257    3.3 Soil organic carbon across pH classes

SOC content is greater in more alkaline soils with pH >5.5 (Fig. S1). The relative contribution
of the explanatory variables to the explained variance in SOC is dependent upon pH (Fig. 1,



Fig. 2). For lower pH classes, MAP explains the largest portion of the response variance of
SOC content, while CEC eff. explains the greatest portion of variance in more alkaline soils
Fig. 3 presents the normalized contribution of MAP, MAT, LAI, percent clay and CEC eff. in
explaining the variance of SOC content. The total amount of variance explained (R squared) by
each independent linear model across pH classes ranges between 0.76 (0-30cm depth, pH 6-
6.5) and 0.11 (30-120 cm depth, pH 5.5-6) (Fig. 4). On average, each pH class contains 125±94
(mean ± SD) forest sites that we could analyze with independent linear models (Fig. 4). The
total amount of variance explained by the linear model runs is independent from the number of
forest sites included in the models.

**4 Discussion**

**4.1 Can the effective cation exchange capacity be used as a proxy for soil**
**organic carbon content?**
Our linear model analysis indicates that throughout 1204 Swiss forest sites, encompassing a
pronounced gradient of climate, geology, soil pH and soil orders, CEC eff. has the largest
explanatory potential of SOC content (35 % of response variance in the complete mineral soil
profile) as compared to other environmental variables (Table 2). Since CEC eff. is an edaphic
property which is intimately associated to both the conditions that shaped the soil and the
current physicochemical conditions in the soil, the significant link between CEC eff. and SOC
hints towards a potential of this edaphic property to be used as an integrative proxy of the
present and future content of SOC. In our evaluation we included a range of climatic, vegetation
and edaphic properties, as these variables are commonly regarded as primary controls of SOC
(Torn et al., 2009). However, we cannot rule out that our analysis could be biased by the lack
of available information on the activity by soil organisms (microbial communities and soil
macro- and meso- fauna) (Jackson et al., 2017). The weak relationship between the vegetation



property we adopted as an indication of plant primary production (the LAI) and SOC content
strongly suggests that the quantity of litter inputs is less important for SOC than
physicochemical properties. This is in agreement with mechanistic studies employing isotopic
tracers, as well as spectroscopic and molecular techniques (Hagedorn et al., 2003; Marschner
et al., 2008; Schmidt et al., 2011). The weak relationship may however also result from the lack
of information on the growth of belowground plant biomass as well as on differing
decomposition rates of above and belowground plant debris before incorporation into the soil
(Hobbie et al., 2010). Nevertheless, the results of our analysis clearly demonstrate that CEC eff.
explains a larger portion of the variance in SOC content than the percentage of clay - on which
many biogeochemical models rely as a proxy of SOC. Such pattern might be partly related to
the contribution of variably charged functional groups of organic matter to CEC eff. but it also
corroborates the emerging conceptual understanding that other soil physicochemical properties
predict the content and potential persistence of SOC better than the amount of clay (e.g.
Rasmussen et al., 2018).

**301     4.2 How is the variance of soil organic carbon content explained by**

**302     environmental variables at different soil depths?**

In agreement with vast ranging observations showing that SOC content decreases with soil
depth (e.g. Jobbágy and Jackson, 2000; Rumpel and Kögel-Knabner, 2011), we observed a
higher SOC content in the upper 30 cm of the mineral soil as compared to that found at lower
depths (Table 2). This pattern is commonly related to a decline in the inputs of plant-derived
organic matter with depth because litterfall from leaves, needles and twigs occurs on the soil
surface, and this material is only partially translocated to the subsoil in dissolved form or via
bioturbation. Moreover, plant root densities are commonly smaller in subsoils than in topsoils
(Jackson et al., 1997). Microbial communities involved in the breakdown of organic matter also
vary across depth profiles (e. g. Lindahl et al., 2007). Our linear model analysis demonstrates





that CEC eff. is the strongest predictor of SOC in topsoils (explaining 60 % of the total variance
in SOC content) while climatic variables (MAT and MAP) are stronger predictors of SOC in
subsoils (Table 2). The primary influence of CEC eff. on SOC content in topsoils as compared
to deeper soil depths can be attributed to the overall larger presence of organic matter that
contributes markedly to a higher number of available negative charges in soils to which cations
can bind. The dominant control of climate on the content of SOC in subsoils is instead
connected to a significant transport of fresh C to deeper soil depths through leaching of
dissolved organic matter (Kaiser and Guggenberger, 2000), which increases with increasing
precipitation and a greater accumulation of organic horizons, especially under cool and humid
climatic conditions (Gosheva et al., 2017). Analyses of the content of SOC in the soil solution
and in the organic debris residing inside and outside aggregates and bound to minerals would
be necessary to test these mechanisms across complete soil profiles in Swiss forest sites.

**4.3 How does soil pH influence the variance of soil organic carbon content**
**explained by different environmental variables?**

Overall, we observed a significant and complex influence of soil pH on the content of SOC
confirming results from previous studies (Rasmussen et al., 2018; Newcomb et al., 2017). Our
study provided a unique soil pH gradient (spanning from 2.8 to 8.1, Fig. S2) which largely
resulted from the complex topography and orogeny of the territory of Switzerland
encompassing various types of parent material, made up of sedimentary, crystalline or
metamorphic rocks with highly variable mineralogical compositions (Gosheva et al., 2017). We
found that SOC content is higher in more alkaline soils with pH >5.5 (Fig. S1). However, it has
to be noted that SOC stocks in Swiss forests are generally not higher in calcareous than acidic
soils (Gosheva et al., 2017), indicating that acidic soils e.g. in the South of Switzerland (Fig.
S2), have a lower SOC content but are generally deeper. Moreover, our analysis revealed that
the relative control of climatic, vegetation and edaphic variables on SOC evolves as a function





of soil pH. MAP explained most of the variance in SOC content at low pH (pH <5.5), while
CEC eff. was the dominant variable at higher pH values (pH >4.5 in topsoils and pH >6 in
subsoils) (Fig. 2).

The strong influence of MAP on SOC that we observed at low pH is likely primarily

related to the reaction of organic ligands with aluminium cations ($Al^{3+}$) that often occurs in
many soils of regions with a high MAP and thus high water availabilities (Chadwick and
Chorover, 2001; Torn et al., 1997; Blaser et al., 1997; Blaser and Sposito, 1987; Eckmeier et
al., 2010). Firstly, in acid soils the presence of $Al^{3+}$ cations is commonly associated to the
production of organo-metal complexes during the process of weathering. Secondly, a strong
chelation of $Al^{3+}$ by organic acids in Swiss forest soils characterized by a high MAP and acid
soils, is pointed out by the high levels of exchangeable aluminium ($Al^{3+}$) found at low pH
classes (Fig. S1), and by the observation of the highest MAP regimes in regions with the most
acidic soils (Fig. S2, S3). These patterns corroborate previous results showing that the relatively
humid climate of the Southern Alps promotes both the formation of pedogenic oxides and the
leaching of dissolved organic C from the organic layer to the mineral soil, where organic matter
i.e. tannins and other polyphenols forms organo-metal complexes (Eckmeier et al., 2010). The
peak in the explanatory power of MAP on the content of SOC at pH 4.5-5 (Fig. 3) is probably
related to the greater occurrence of Luvisols and Podzols in this pH range (Fig. S4). These soil
orders are characterized by translocation of clay and organic matter.

The dominant contribution of CEC eff. in explaining the variance of SOC in higher pH

classes can be explained by the established knowledge that the amount of available negative
charges to which cations can be bound on organic matter - together as on allophane and some
1:1-type clays - increase with rising pH (Weil and Brady, 2016). In particular, the deprotonation
of carboxyl and hydroxyl groups present in organic matter has often been observed to contribute
markedly to the greater number of available negative charges in soils, and more so at higher pH
values (Krishnaswamy and Richter, 2002; Tate and Theng, 1980; Sullivan et al., 2006).



Furthermore at pH levels higher than 6 the charges of 2:1-type clays have been observed to
increase due to a deprotonation of exposed hydroxyl functional groups (Tournassat et al., 2016).

**5 Conclusions**
In conclusion, our study of Swiss forest soils provides a first indication that CEC eff. could be
used as an integrative proxy of SOC content and its potential preservation. Determining whether
CEC eff. is an effective proxy for the content of SOC also in non-forested ecosystems and in
other geographical locations would be a further step forward to test its broad applicability in
Earth System models. This information could be compiled starting from available soil profile
samples, soil surveys and monitoring programs at the country, continental or climatic zone level
(e.g. http://icp-forests.net/, http://ncsslabdatamart.sc.egov.usda.gov/). Some global gridded
data about CEC. eff has also already been derived (Hengl et al., 2017) by the international soil
reference and information center ISRIC. We observed that CEC eff and SOC are closely linked
especially in the top mineral soil (0-30 cm depth) as compared to deeper depths. This is due to
the overall larger presence of organic matter contributing to a higher number of available
negative charges in topsoils. In subsoils most of the variance in SOC is instead explained by
climate, which in our study is potentially related to increasing weathering and transport of fresh
C to greater soil depths through leaching of dissolved organic matter with increasing MAP. Soil
pH strongly affects the explanatory power of CEC eff. on the content of SOC in Swiss forest
soils, with CEC eff. being the dominant variable at pH >4.5 in topsoils and pH >6 in subsoils.
Since 73% of the land surface is covered by soils with pH >5.5 (area calculated according to
the global pH dataset IGBP-DIS (1998)) the consideration of CEC eff. as an integrative proxy
for the potential alterations of SOC content could strengthen our ability to predict future
changes in the belowground C reservoir.





## Author contributions


MWIS, SZ, FH conceived the idea of this analysis, SZ and LW provided and quality checked
the data, EFS developed the ideas for this manuscript and considering the preliminary analyses
by VW defined the final statistical approach. EFS prepared and wrote the manuscript with
contributions from all co-authors.

## Acknowledgements


Evaluations were partly based on data from the Swiss Long-term Forest Ecosystem Research
programme LWF (www.lwf.ch), which is part of the UNECE Co-operative Programme on
Assessment and Monitoring of Air Pollution Effects on Forests ICP Forests (www.icp-
forests.net). We are in particular grateful to Peter Jakob for his contribution to the soil database,
and the technical staff of the WSL research unit Forest Soils and Biogeochemistry for collecting
and analyzing the soil samples. This study was financed by the Swiss National Science
Foundation SNSF project 'Deep C' (number 172744) and the University Research Priority
Program for Global Change and Biodiversity of the University of Zurich. Data supporting this
paper is provided in a separate data Table.

## Competing interests


The authors declare no competing interests.

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





**Figures:**

**Fig. 1:** Pearson's correlations between the content of SOC and explanatory variables for (a) complete mineral soil profiles (0-120 cm depth), (b) topsoils (0-30 cm depth), (c) subsoils (30-120 cm depth). MAP, mean annual precipitation (mm), MAT, mean annual temperature (°C), LAI, leaf area index, clay, percentage of clay (%), CEC, effective cation exchange capacity (mmolc kg$^{-1}$). Those correlations with $P > 0.05$ are considered not significant (NS) based on Bonferroni's correction.

**Fig. 2:** Amount of explained variance by climatic, vegetation and edaphic variables in predicting the content of SOC (g kg$^{-1}$) across pH classes for (a) complete mineral soil profiles (0-120 cm depth), (b) topsoils (0-30 cm depth), (c) subsoils (30-120 cm depth). Horizontal bars show the amount of variance explained by each variable (R squared) in the linear models as calculated using the 'lmg' metric in the R package 'relaimpo'. Significant variables (p-value <0.05) are marked by an asterisk (*). MAP, mean annual precipitation (mm), MAT, mean annual temperature (°C), LAI, leaf area index, clay, percentage of clay (%), CEC, effective cation exchange capacity (mmolc kg$^{-1}$). The total amount of variance explained (R squared) by each independent linear model is presented in Fig. 4 together with the number of samples used.

**Fig. 3:** Relative contribution (percentage of R squared) of climatic, vegetation and edaphic variables in predicting the content of SOC across pH classes, for (a) complete mineral soil profiles (0-120 cm depth), (b) topsoils (0-30 cm depth), (c) subsoils (30-120 cm depth). The response variance explained by each environmental variable was normalized to sum to 100 %. MAP, mean annual precipitation (mm), MAT, mean annual temperature (°C), LAI, leaf area index, clay, percentage of clay (%), CEC, effective cation exchange capacity (mmolc kg$^{-1}$).

**Fig. 4:** (a) Number of samples and (b) total variance explained (R squared) by each independent linear model as presented in Fig. 1.



**Fig. 1:**

| (a) Complete mineral soil profile 0 – 120 cm depth | | | | | | | | | |
|---|---|---|---|---|---|---|---|---|---|
| pH | <4 | 4-4.5 | 4.5-5 | 5-5.5 | 5.5-6 | 6-6.5 | 6.5-7 | 7-7.5 | >7.5 | All pH |
| MAP | 0.27 | 0.53 | NS | NS | NS | NS | NS | NS | NS | 0.21 |
| MAT | -0.21 | NS | NS | -0.52 | NS | NS | NS | NS | NS | -0.13 |
| LAI | -0.22 | NS | NS | NS | NS | NS | NS | NS | NS | -0.20 |
| Clay | -0.23 | -0.25 | NS | NS | NS | NS | NS | NS | NS | NS |
| CEC | 0.19 | NS | NS | NS | NS | 0.78 | 0.86 | 0.84 | 0.62 | 0.51 |
| (b) Topsoil 0-30 cm depth | | | | | | | | | |
| pH | <4 | 4-4.5 | 4.5-5 | 5-5.5 | 5.5-6 | 6-6.5 | 6.5-7 | 7-7.5 | >7.5 | All pH |
| MAP | 0.39 | 0.52 | NS | NS | NS | 0.52 | NS | 0.29 | NS | 0.22 |
| MAT | -0.12 | NS | -0.40 | NS | NS | NS | NS | NS | NS | -0.14 |
| LAI | -0.19 | NS | NS | NS | NS | NS | NS | -0.27 | NS | -0.20 |
| Clay | -0.16 | NS | NS | NS | NS | NS | NS | NS | NS | 0.16 |
| CEC | 0.23 | NS | 0.57 | 0.73 | 0.88 | 0.87 | 0.84 | 0.88 | 0.80 | 0.60 |
| (c) Subsoil 30-120 cm depth | | | | | | | | | |
| pH | <4 | 4-4.5 | 4.5-5 | 5-5.5 | 5.5-6 | 6-6.5 | 6.5-7 | 7-7.5 | >7.5 | All pH |
| MAP | NS | 0.42 | 0.43 | NS | NS | NS | NS | 0.29 | 0.21 | 0.21 |
| MAT | -0.45 | NS | NS | NS | NS | NS | NS | NS | NS | -NS |
| LAI | -0.29 | NS | NS | NS | NS | NS | NS | -0.31 | NS | -0.16 |
| Clay | -0.26 | -0.23 | -0.43 | NS | NS | NS | NS | 0.37 | NS | NS |
| CEC | NS | NS | -0.47 | NS | NS | 0.68 | NS | 0.79 | 0.73 | 0.73 |

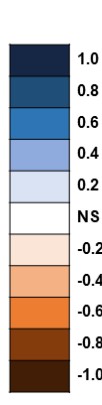

1.0
0.8
0.6
0.4
0.2
NS
-0.2
-0.4
-0.6
-0.8
-1.0





**Fig. 2:**

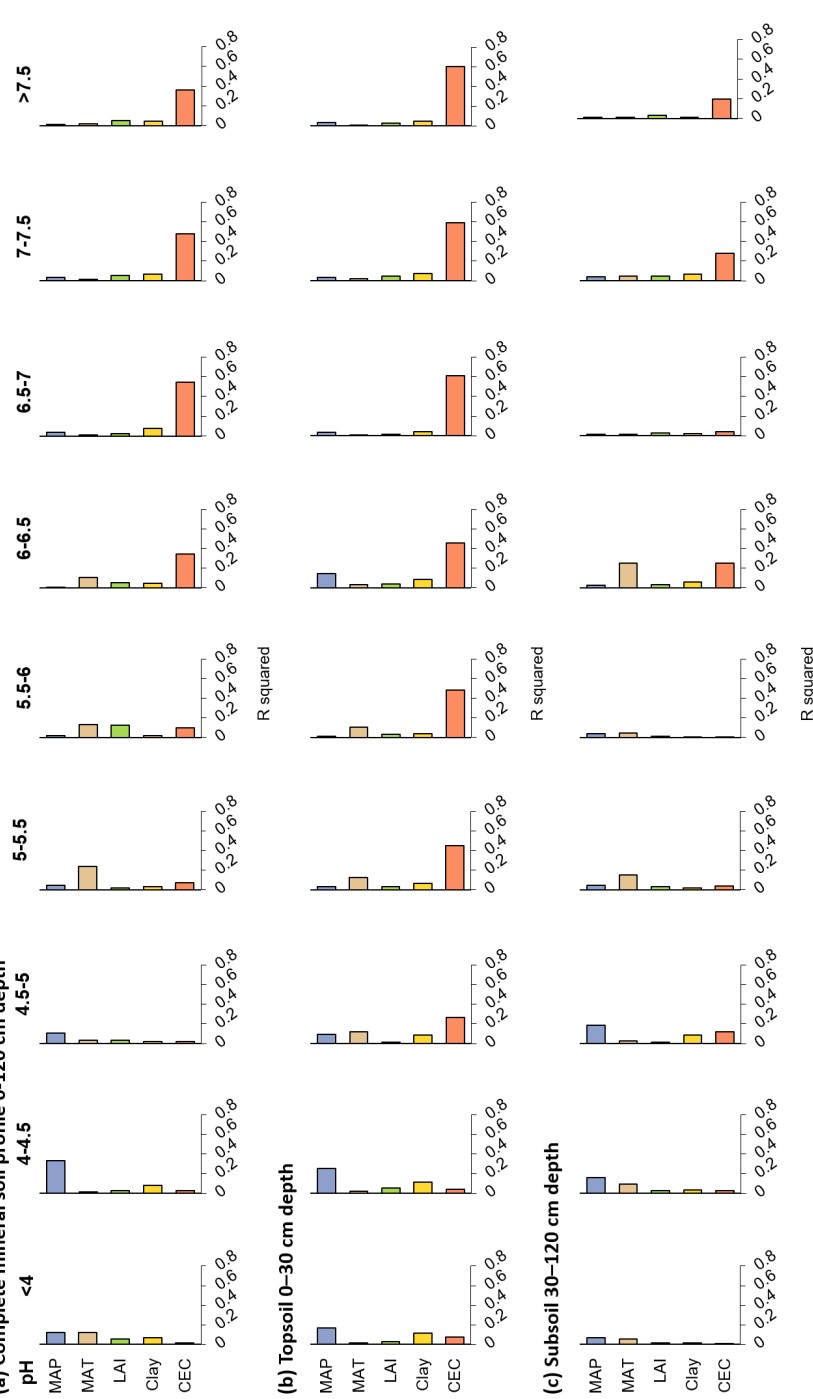



**Fig. 3:**

**(a)**

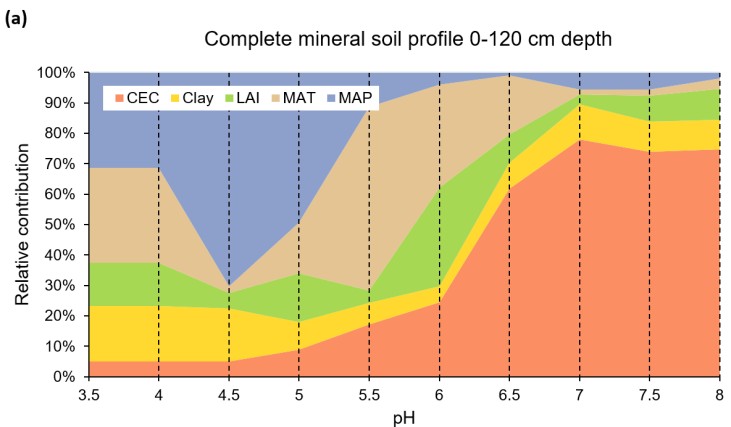

**(b)**

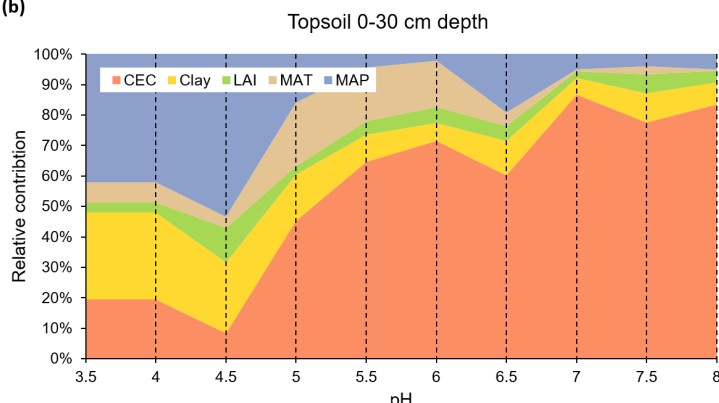

**(c)**

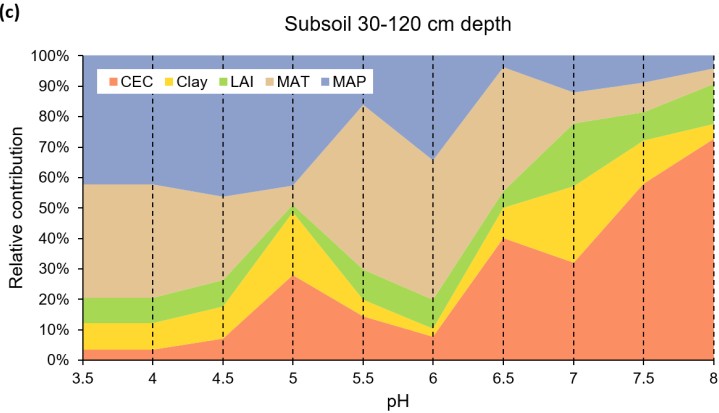





**Fig. 4:**

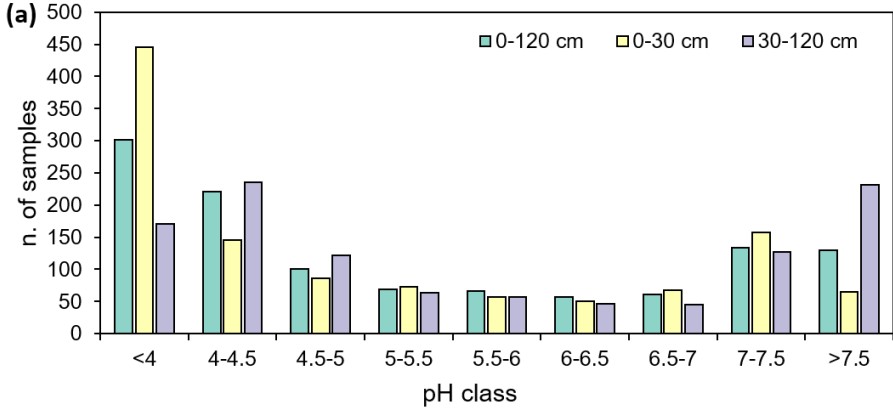

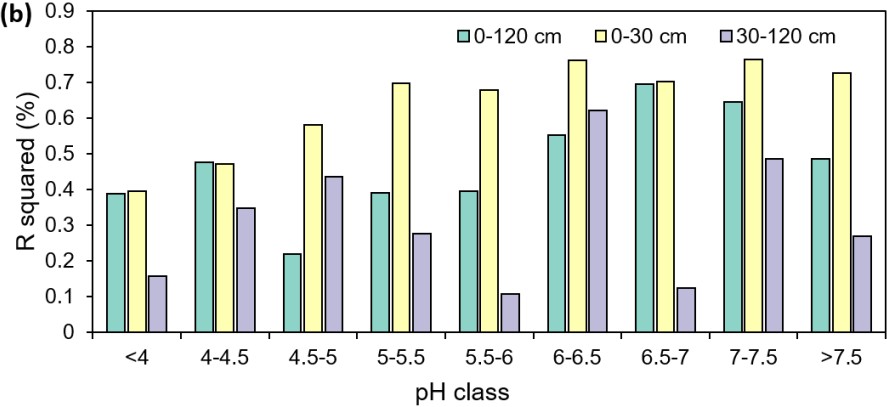





**Table 1:** Range of climatic, vegetation and edaphic variables for complete mineral soil profiles (0-120 cm depth), topsoils (0-30 cm depth) and subsoils (30-120 cm depth). MAP, mean annual precipitation; MAT, mean annual temperature; LAI, leaf area index; SOC, soil organic carbon content, CEC eff., effective cation exchange capacity.

| Soil depth | 0-120 cm | 0-30 cm | 30-120 cm |
|---|---|---|---|
| MAP (mm) | 636-2484 | | |
| MAT (°C) | 0.1-12.0 | | |
| Elevation (m a.s.l.) | 277-2207 | | |
| LAI | 2-7.2 | | |
| SOC (g kg$^{-1}$) | 1.3-376.8 | 6.0-376.8 | 0-229.0 |
| CEC eff. (mmolc kg$^{-1}$) | 11.4-753.2 | 20.4-1046.4 | 6.2-727.9 |
| Clay (%) | 1.2-74.7 | 1.5-74.6 | 0.8-75.6 |
| pH | 3.0-8.0 | 2.8–8.0 | 3.0-8.1 |



**Table 2:** (a) Standardized coefficients indicating explanatory power of climatic, vegetation and edaphic variables in predicting the content of SOC, and (b) amount of variance (% of R squared) explained by the explanatory variables for complete mineral soil profiles (0-120 cm depth), topsoils (0-30 cm depth) and subsoils (30-120 cm depth). The R squared of metrics is normalized to sum to 100 %. MAP, mean annual precipitation; MAT, mean annual temperature; LAI, leaf area index; CEC eff., effective cation exchange capacity; n., number of samples; NS, non significant.

| | Complete pH range | | |
|---|---|---|---|
| **Depth (cm)** | **0-120** | **0-30** | **30-120** |
| **(a) Standardized regression coefficients in predicting SOC (g kg$^{-1}$)** | | | |
| **MAP (mm)** | 0.31 | 0.30 | 0.26 |
| **MAT (°C)** | -0.14 | NS | -0.18 |
| **LAI** | -0.16 | -0.13 | -0.11 |
| **Clay (%)** | -0.26 | -0.26 | -0.23 |
| **CEC eff. (mmolc kg$^{-1}$)** | 0.48 | 0.66 | 0.32 |
| **R squared** | 0.26 | 0.39 | 0.17 |
| **n.** | 1138 | 1146 | 1099 |
| **(b) Relative importance (% of R squared) in predicting SOC (g kg$^{-1}$)** | | | |
| **MAP (mm)** | 26 | 18 | 31 |
| **MAT (°C)** | 15 | 4 | 28 |
| **LAI** | 17 | 9 | 7 |
| **Clay (%)** | 7 | 9 | 17 |
| **CEC eff. (mmolc kg$^{-1}$)** | 35 | 60 | 17 |