# Peer review of "Is the content and potential preservation of soil organic"

_Biogeosciences, 2019_

## Short Comment (SC1) · 4 Mar 2019

Solly et al. (2019) argue that the effective cation exchange capacity (CEC_eff) could be used as a proxy for the potential preservation of SOC. They further argue that the derived preservation potential of SOC could be beneficial for SOM modelling and replace the commonly used clay or clay + silt proxies for the preservation of SOM related to mineral association.

The authors correctly mention that CEC is directly influenced by the amount of organic matter in soils (line 102 and line 378). To my mind it could make sense to try to distinguish between the CEC (CEC_clay) that is provided by clay size minerals and the CEC

provided by organic matter (CEC_OM). However, if I understood the Methods sections correctly, the CEC measured for this paper is indeed the overall CEC. Compare for example with Klamt and Sombroek (1988) and how they tried to distinguish between CEC_clay and CEC_OM.

Therefore, it should not come as a surprise that SOC and CEC are highly correlated in the topsoil, since CEC is determined to a large part by the presence of SOC. In my opinion, the use of CEC as proxy for the preservation potential of SOC in soils is circular and empirical relationships of SOC with CEC should not be used in modelling to parameterize a preservation capacity.

References

Klamt E, Sombroek W (1988) Contribution of organic matter to exchange properties of Oxisols.

Solly EF, Weber V, Zimmermann S, Walthert L, Hagedorn F, Schmidt MWI (2019) Is the content and potential preservation of soil organic carbon reflected by cation exchange capacity? A case study in Swiss forest soils. Biogeosciences Discussions, 1-32.

---

## Short Comment (SC2) · 6 Mar 2019

Thank you! We realize that we have to clarify the purpose of our exercise. Our intention was not to describe a mechanistic relationship. We rather propose to explore if CEC eff. can act as an extensively available "integrative proxy" to substitute for those variables probably controlling SOC stabilization but which are difficult or impossible to quantify for large areas. If successful, this integrative proxy could allow to produce large scale SOC inventories, but if CEC eff. could help to improve mechanistic soil models remains an open question (this will be specified in lines 41-24, and 285-387).

To briefly summarize our arguments in the introduction sections (lines 70-126): Indeed

it is well known that in the topsoil SOC and CEC are highly correlated and mechanistically linked (i.e. SOC contributes to CEC). Fortunately, SOC is a variable that can be measured directly. But other variables important for SOC preservation potential are more difficult to measure, especially for large areas. Those variables require methods that can be too costly or time-consuming for large sample sets (quantification of clay content, clay mineralogy, specific surface area, Al-, Fe- organo-metal complexes), or robust methods for quantification do not yet exist (e.g. quantification of short range order minerals). CEC eff. however, could act as an "integrative proxy" for all of the variables mentioned before. CEC eff. is an edaphic property which is intimately associated to both the conditions that shaped the soil and the current edaphic physicochemical conditions. And CEC eff. is measured routinely to assess soil fertility for agricultural and forest use, and a wealth of data of past and present CEC eff. already exists.

---

## Short Comment (SC3) · 6 Mar 2019

In the published version of SC2 'CEC eff. as "integrative proxy" potentially helps to improve inventories, not mechanistic models' line numbers were incorrectly given as '41-24, and 285-387'; the correct line numbers are '41-42 and 385-387', at the end of the abstract and end of the conclusions respectively.

---

## Referee Comment (RC1) · Anonymous Referee #1 · 1 Apr 2019

The current study presents a thoughtful consideration of factors affecting SOM abundance across a dataset of considerable size and quality. The examination of variance in these relationships with depth is especially interesting, and stands to lend useful and insightful information for soil C cycle modeling efforts. This dataset is extremely valuable, and I believe the authors will be able to extract some very meaningful conclusions from this work. The manuscript is well written. The introduction could benefit from a reread by the authors and some slight revision for clarity, but the main concepts being discussed are timely and well-articulated for the most part.

The language surrounding the concept of proxy variables is used inconsistently

throughout the manuscript. In the introduction, the authors hypothesize that CEC-eff can be used as an integrative proxy, representing the sorptive capacity associated with reactive soil surfaces (including organic surfaces, which presents some logic problems). In other places, the authors suggest that CECeff could be used as an integrative proxy of SOC content and its potential preservation. If CECeff is an integrative proxy of stabilization mechanisms, then it would be a predictor of SOC content. If CECeff is an integrative proxy of SOC content and stability, then it would potentially be used instead of SOC in models. By definition, a proxy is, "...a measurement of one physical quantity that is used in the place of a different quantity that would be too difficult or expensive to measure directly" (Bailey et al., 2017).

I believe the biggest issue the authors must effectively address during revision is the choice of CECeff as their explanatory variable of choice. SOM often accounts for a very large portion of the overall CECeff of a particular soil sample. Therefore, CECeff is dependent on SOM content, not the other way around as the model in the paper suggests (using CECeff as an explanatory variable in a model for SOC). That isn't to say that CECeff couldn't be couldn't be used as a proxy for SOC content, but it would seem more effective to just measure SOC content since CECeff only exhibits a moderate correlation with SOC and is just as laborious to measure. The dependence of CEC on SOM content would explain why correlations among CECeff and SOC are stronger in surface soils where SOC is more abundant, as is stated in the discussion. To some extent, the same argument could be made against the findings of Rasmussen et al., 2018, since exchangeable Ca comes from organic exchange sites not associated with mineral surfaces as well as from organo-mineral cation bridging. I believe the use of exchangeable Ca is somewhat more defensible since its role in SOM stabilization is understood on a mechanistic level. It forms cation bridges between organic and inorganic surfaces through ligand exchange. Monovalent cations do not form cation bridges, and therefore cannot contribute to the stabilization of SOM. Exchangeable Mg does not lend itself to stable cation bridges due to a smaller ionic radius. The authors will have to justify from a mechanistic perspective, how CECeff functions to promote

SOM accumulation and/or stability.

The introductory material suggests that CECeff might act as an effective integrative proxy for properties such as surface area, short-range-order mineral content, clay content and soil organic matter. Here again, is a circular argument. The authors are stating that variance in reactive mineral surface area and SOM exchange sites can be predicted by changes in CEC. They then claim that changes in SOC can be predicted by changes in CEC. We all know that SOM and SOC are inherently linked, and CEC is highly dependent on SOM, so why bother with the proxy? Just measure SOM, which will basically give you a SOC value. Also, explanatory variables included in soil C models must have predictive capacity in order to be useful. We need explanatory variables that will be able to predict how SOC stocks will change in abundance or stability. Because CEC is so heavily influenced by SOM concentration, it changes as a result of changes in SOM concentration, not the other way around. Yes, they are correlated to some degree, but I believe CEC is the dependent variable and SOM (and therefore also SOC) is the explanatory variable.

Also, if the desire is to prove that CEC can be used as an integrative proxy for stabilization mechanisms, then the wrong model has been constructed. In order to prove that CEC is accounting for variation in oxalate-extractable metals, clay content, and surface area, a model would have to be constructed with CEC as the dependent variable, and oxalate-extractable metals, clay, surface area, etc. as the explanatory variables. It seems like the first hypothesis of this paper should be, "CECeff serves as an effective integrative proxy for variables such as metals, clays, and surface area". The authors would then prove the correctness of that hypothesis by using a statistical model to link variation in CEC with variation in metals, clays, and surface area. Then the argument would follow that CEC is much easier to measure than these other properties, as stated in the introduction, and the second hypothesis would then follow, "Because CEC is an effective integrative proxy of SOM stabilization mechanisms, CEC can be used to predict changes in the stability and abundance of SOC". Then a model similar to the one

currently presented would be appropriate.

I believe the modeling work in the manuscript could be improved by a slightly different statistical approach. It doesn't seem appropriate to test for the significance of the explanatory variables for the 0-120 cm models without taking depth into account. Depth is a confounding variable due to the fact that most soil physicochemical characteristics vary predictably with depth. The chosen approach then was to split surface soils from subsurface soils (0-30 cm and 30-120 cm) to examine how the relative influence of difference explanatory variables varied with depth. I believe a more appropriate approach would be to apply a linear mixed model using all the explanatory variables as fixed effects. Depth and all its interaction terms would also be included as fixed effects, with SOC as the dependent variable. The resulting model would indicate which of the climatic or physicochemical variables varied in their influence with depth (which fixed effects were significant). The two-way interaction terms that are significant should be fairly easy to interpret given how the data has been transformed. I'm also confused about why pH and its possible interaction terms were not tested for significance. One of the main findings of the paper is that the relative importance of explanatory variables depend on pH. Perhaps pH could be a more useful proxy than one or more of the other variables currently used in the model? Was pH ever included in a model? Are there other soil physicochemical properties that the authors could explore in lieu of CECeff?

I would also ask the authors to explain their choice of environmental parameters. Why use LAI instead of NPP? Many soil scientists would argue that NPP would be a better predictor of OM inputs to the soil. Why use MAT and MAP instead of PET or a soil moisture regime index? The authors indicate that differences in moisture are important regulators of the downward propagation of C in these soils, because of differences in leaching depth. PET and/or a soil moisture index would do a better job of representing leaching potential because the seasonality and form of precipitation matters, not just the total amount of precipitation.
* * *

---

## Referee Comment (RC2) · Anonymous Referee #2 · 7 Apr 2019

This MS is succinct and well written and uses a robust data set to test their central question. There appears to be a critical flaw in this analysis – organic matter is likely contributing the majority of CEC in a large number of the samples.

CEC at low pH is dominated by permanent charge on clay minerals but as pH rises variable charge on clay and organic matter take over as the dominant control on CEC. I think the results presented in Fig 1 and 3 are a nice demonstration of this phenomena. Given the large range in OM in these samples, it is highly probable that variable charge of OM is driving the correlations seen in this analysis.

Unless there is a way of removing the confounding influence of OM on CEC espe-

cially at more neutral to basic pH levels, I do not see how this study is publishable. Alternatively, the authors can provide a very convincing argument as to why the biogeochemistry community should accept the findings. Perhaps I'm just dense and don't get it, but, if that is the case, then the authors need to spend some time in the introduction laying out the logic behind how CECeff is not confounded by OM content in the context of this analysis.

A few other comments: L40 Does correlation with a physiochemical property mean more potential for preservation?

L113-119 Measurement of Al and Fe forms is equally laborious as calculating CEC (both involve similar extraction protocols then quantification on an ICP or AAS or similar), so this argument is a bit of a red herring

L160 Exchangeable cations measured from an unbuffered solution will overestimate Ca in calcareous soils and overestimate Na in sodic soils (probably aren't any in Swiss forests), therefore CEC calculated by summing cations instead of by further displacement of the NH4 is not reliable for these soil types.

L238 Is mmolc/kg an acceptable unit? I thought ccmolc/kg was the standard

---

## Referee Comment (RC3) · Anonymous Referee #3 · 14 Apr 2019

Overall, this is an important analysis of proxies that can predict SOC content. This manuscript fits well with recent syntheses and reviews like Rasmussen et al. 2018 and Rowley et al. 2018 (both in Biogeochemistry). The authors tested the ability of CEC, clay, LAI, MAT, and MAP to predict the weighted average SOC content in the surface, subsurface, and whole soil profile in >1000 forest soil profiles across Switzerland. They found that effective CEC was the best predictor of SOC content at higher pH's in the whole soil profile and surface 30 cm of the soil profile while MAP was a stronger predictor at lower pH in the whole profile and surface soil. For the subsoil, both climate variables (MAT and MAP) were the strongest predictors likely due to greater weathering and leaching of organic molecules through the soil profile.

[Figure]

The statistics are sound and clearly presented. The figures are clear, though figures 2 and 3 are a bit redundant in that they show the same trends. I think Figure 3 is clearer and suggest only using that figure, but given this is an online open access journal, I see no harm in including both if the authors feel strongly about including both.

My largest comment is that I wonder how single cations would be as predictors, such as Ca and Fe or Al as in Rasmussen et al. 2018 as it would be interesting to test those findings with a different dataset. For example, I wonder how much the strong CEC relationship at higher pH is driven by the Ca cations alone and whether Al ions would better explain the MAP relationship at low pH. To test individual cations would likely illuminate mechanisms better and they may even have stronger relationships with SOC than CEC, but individual cations would then not be the integrative proxy that the authors are seeking.

Lastly, I am not sure if it belongs in the Introduction or Discussion, but Rowley's 2018 synthesis, "Ca-mediated stabilization of organic carbon" should be cited in this paper as it also touches upon pH differences in the controls on SOC content. Figure 3 is particularly relevant.

I have some minor comments where the manuscript needs some clarification and where the findings of Rasmussen et al. could be more accurately presented. Specific comments:

Abstract L23: delete "as compared to the mere quantification of clay-sized particles" because as you stated in the intro, that is not a trivial analysis to do. Introduction: L79: Please clarify "exchangeable Ca and Fe, and Al oxyhydroxides". Rasmussen et al. tested the predictive capabilities of oxalate extractable Fe and oxalate extractable Al. These are measures of organo-metal complexes and short range order minerals, not exchangeable Fe or all Al oxyhydroxides. L83: Add that soil pH also determines the relative charges of organic molecules and soil minerals and thus the likelihood that organic molecules will sorb to minerals, so not just organo-metal complexes with

ligands. L84: The line about depth should be a separate sentence as it is unclear here what you want to emphasize about soil depth. L114-116: I disagree that Fe and Al variables cannot be measured on large datasets. Doesn't the analysis in Rasmussen et al. of a large soil dataset, which you cite for this sentence, contradict that statement?

Methods: L147: please define "fine earth" does this mean, you corrected for rocks? How did you classify "fine"? L185: What time period during the growing season was used to determine LAI? L218: The line "for each statistical test $P<0.05$ was . . ." directly contradicts the previous statement. Maybe write, "For all other statistical tests . . ."

Results: When reading the results, my first question was what the distribution of samples among the different pH classes were. You might want to move up Figure 4 to the beginning as it strengthens the interpretation of your analyses to know how evenly distributed the samples were among pH classes.

Discussion: Line 299: Here is a good place to cite Rowley et al. 2018. L314-317: This statement here leads to a bit of a 'chicken and egg' conundrum. Is there more Ca because negative charges on OM can bind to it or is there more SOM because there is more Ca to bind to it? Maybe it doesn't matter for predictions. I have the same issue for when pyrophosphate extractable Fe and Al are used to predict SOC as that extraction targets organo-mineral complexes. Maybe the Ca stabilization mechanisms brought up by Rowley could help here. L317: Tone down this statement to "may be instead" in place of "is instead" as to know for sure you would need a mechanistic test as you nicely point out below. L367-360: Please reword this sentence. I found this sentence to be confusing in its structure, particularly the clause in dashes, and had to reread it several times.

---

## Referee Comment (RC4) · Anonymous Referee #3 · 14 Apr 2019

Just realized I was really thinking too much about Ca in my comment! By the chicken and egg conundrum I meant to say CEC, which is as reviewer 2 pointed out, confounded by SOM. Unlike reviewer 2, I don't think this makes it unpublishable but it may be worth looking at Ca only as a proxy as well to help sort it out!

---

## Author Comment (AC1) · 23 Aug 2019

We thank the three anonymous referees for their comments about our work. We agree with most of their comments and added an extra dataset, which we additionally analyzed to address their concerns. Below, we provide a detailed response to each of their comments. The new dataset consists of a subset of soil profiles from the original dataset where iron and aluminum oxides where measured by oxalate extraction in each of the horizons within a soil profile.

In addition to thoroughly addressing the comments of the referees, we propose to clarify the change of focus resulting from the re-analysis of the data, by adding the prefix \*sub\* to the title: "Is the content and potential preservation of subsoil organic carbon reflected by cation exchange capacity? A case study in Swiss forest soils"

Anonymous Referee #1

The current study presents a thoughtful consideration of factors affecting SOM abundance across a dataset of considerable size and quality. The examination of variance in these relationships with depth is especially interesting, and stands to lend useful and insightful information for soil C cycle modeling efforts. This dataset is extremely valuable, and I believe the authors will be able to extract some very meaningful conclusions from this work. The manuscript is well written. The introduction could benefit from a reread by the authors and some slight revision for clarity, but the main concepts being discussed are timely and well-articulated for the most part.

We thank the referee for the careful analysis of our paper and for the constructive comments from which the manuscript will largely benefit. While addressing the comments raised by the referee, regarding both the main concepts being discussed and the analysis of the dataset, we plan to rewrite considerable parts of the manuscript. Please see the response to each of the comments below, where we provide detailed information on the main corrections we intend to make.

The language surrounding the concept of proxy variables is used inconsistent throughout the manuscript. In the introduction, the authors hypothesize that CEC-eff can be used as an integrative proxy, representing the sorptive capacity associated with reactive soil surfaces (including organic surfaces, which presents some logic problems). In other places, the authors suggest that CEC-eff could be used as an integrative proxy of SOC content and its potential preservation. If CEC-eff is an integrative proxy of stabilization mechanisms, then it would be a predictor of SOC content. If CECeff is an integrative proxy of SOC content and stability, then it would potentially be used instead of SOC in models. By definition, a proxy is, "...a measurement of one physical quantity that is used in the place of a different quantity that would be too difficult or expensive to measure directly" (Bailey et al., 2017).

We agree that the language surrounding the concept of proxy variables was not used correctly. We intend to clarify this inconsistency by specifying that this study aims to test whether the effective cation exchange capacity (CEC eff.) could be used to

describe the potential preservation of soil organic carbon (SOC) in Swiss forests and finally be used to feed models. It is not about measuring CEC eff. instead of SOC.

This is certainly only reasonable for soils profiles or depth increments where soil organic matter contributes little or nothing to the overall CEC eff.

The contribution of soil organic matter (SOM) to CEC has for instance been shown by Parfitt et al. (1995) to be greater in surface horizons than at deeper soil depth for several different soil orders at pH 7. Moreover, Helling et al. (1964) clearly indicated that the contribution of the charge of soil organic matter depends on the pH of the soil, with a greater mean relative contribution of organic matter to CEC increasing from pH 2.5 to pH 8. Considering these observations, it seems plausible to hypothesize that CEC eff. might be a possible predictor of SOC in subsoils falling in specific ranges of pH values in which SOM contributes little or nothing to the CEC eff.

Very recently, several pioneering attempts to estimate SOC storage potential have been published. Besides climate variables, they showed that soil chemical and physical properties (including pH, exchangeable base cations, base saturation..) where primary drivers of SOC dynamics (e.g. Chen et al., 2019; Luo et al., 2017; Doetterl et al., 2015; Luo et al., 2019; Vos et al., 2019). The advantage of using variables such CEC eff. and pH is that these are measured routinely for agricultural and forest management, and rich datasets have been collected in local, regional and national databases. Thus, we think that testing if CEC eff. could serve as a predictor for potential preservation of subsoil SOM - also under a future, changing climate- could be a timely contribution.

Below we provide 1) a mechanistic explanation behind the hypothesized relationship that we plan to test, and 2) information on how we can address the confounding effect of organic surfaces.

1) Mechanistic explanation of how CEC eff. can influence SOC preservation:

We think it makes sense to test whether CEC eff. can play a role in describing the preservation of SOC due to the well established knowledge that SOC can be stabilized through interactions with metal cations. These interactions occur through the formation of polyvalent cation bridges between organic molecules and minerals and/or other organic compounds in soil. The stability of SOC is increased thanks to these interactions by its reduced chance of being degraded by degrading enzymes through the transfer of dissolved soil organic matter to solid phase. Moreover, flocculation and precipitation of dissolved organic matter by cations can occur. For instance, the flocculation of organic matter has been observed to be dominated by the cation $Al^{3+}$ in acidic soils. Rowley et al. (2018) have recently explained that due to the ability of $Ca^{2+}$ to form efficient outer sphere bridge units, this cation is also a fundamental flocculating agent in soils.

Since the CEC eff. is a measure of the quantity of exchangeable cations that are retained through sorption on reactive soil particle surfaces, it represents a sum of the activity of positively charged metal cations in soils and the exchangeable hydrogen-ion acidity. Although CEC eff. is thought to be

primarily related to outer-sphere adsorptive reactions, recent advances are arguing that some cations may additionally readily exchange their hydration shells and create inner-sphere complexes with organic compounds (Rowley et al., 2018). The amount and activity of cations commonly depends on cation specific characteristics (ionic potential, polarizability of their electron cloud, propensity to retain their hydration shell) and variations with environmental conditions (including pH, temperature, pressure, soil solution composition etc…). For instance, $Fe^{3+}$ normally forms insoluble precipitates for most environmentally relevant pH conditions and is rarely found as free ion in soils. $Al^{3+}$ is instead usually observed to play a relevant role in forming outer sphere cation exchange and inner sphere ligand exchange in acidic soils. Base cations such as $Ca^{2+}$, $Mg^{2+}$ $K^+$ and $Na^+$ are mainly thought to behave as exchangeable cations. However, like $Al^{3+}$ some base cations such as $Ca^{2+}$ are weakly polarizable and tend to form O-containing ligands through ionic bonding, indicating that they may theoretically be able to form inner sphere complexes (Sposito, 2008; Rowley et al., 2018). The actual affinity of each cation to inner-sphere exchange sites is very complicated to study due to the multitude of the organic ligands present in soils and by behavior of cations changing with environmental conditions (e.g. pH, temperature, pressure etc…). While monovalent cation $Na^+$ does not participate in inner sphere complexes, and $K^+$ only participates in these complexations in interlayers of certain phyllosilicates (Rowley et al., 2018), modelling studies have indicated the potential of $Ca^{2+}$ to interact with SOC through both inner- and outer-sphere processes  (Minick et al., 2017). Sutton et al. (2005) have for instance found that the complexation of deprotonated carboxyl groups with $Ca^{2+}$ where predominantly derived from inner-sphere complexation during a modelling exercise. Regarding the interactions of $Ca^{2+}$ and SOC, Rowley et al. (2018) have additionally pointed to the consideration that due to the distinctively high reversibility of outer sphere interactions, the widely established correlation between exchangeable $Ca^{2+}$ and SOC is likely not solely attributed to outer sphere processes. The mechanisms leading to a potential role of $Ca^{2+}$ in forming inner-sphere bridging, and hence stabilizing SOC, however remains to be tested.

Overall, considering the above mentioned mechanisms we think that it is plausible that CEC eff. could reflect the potential preservation of SOC. This is certainly only to be tested for soils profiles or depth increments where soil organic matter contributes little or nothing to the overall CEC eff, such as subsoils.

2) Assessment of the confounding effect of organic surfaces on CEC eff:

To separate the contribution of organic matter and inorganic components such as clay minerals to CEC eff. of soils, statistical approaches have been advantageously adopted over the selective removal of each component by chemical treatment (e.g. Klamt and Sombroek (1988)). The reason being that the chemical treatment can cause interference on the derived charges.  We plan to use the partial regression coefficient values and the content of soil organic matter, clay and CEC eff. to distinguish how much of the CEC eff. is driven by soil organic matter and clay minerals. We propose to use the following equation (adapted from Klamt and Sombroek (1988)):

X <- X2 / X1 (X2 = amount of SOM in soil (SOM=SOC in our dataset), X1 = amount of clay in soil)
Y <- amount of CEC eff. in soil/ X1
fit <- lm(Y ~ X)
a <- fit$coeff[1] (CEC of 1 unit of clay in soil)
b <- fit$coeff[2] (CEC of 1 unit of SOM in soil)

Contribution of SOM to soil CEC-eff. =
b * total amount of SOM in soil/ total amount of CEC.eff in soil * 100

The preliminary results are presented in Figure 1 and show that in topsoils SOM contributes between 35% and 50% to CEC eff. of the soil and hence that CEC eff. cannot not be used as a predictor of SOC preservation for surface soil horizons. However, the contribution of SOM to CEC eff. in subsoils was lower, ranging between 0 and 11% in soils with pH lower or equal 6.5 and between 17 and 34% for soils with higher pH. Indicating that CEC eff. could be potentially used as a predictor of SOC content in subsoils of Swiss forests, particularly those soils with a pH lower or equal to 6.5.

**Topsoil 0-30 cm depth**

[Figure]

**Subsoil 30-120 cm depth**

[Figure]

Figure 1: Relative contribution of SOM and Clay to CEC eff. across pH classes of Swiss forest soils. The contribution of SOC and Clay was estimated using partial regression coefficient values and the content of soil organic matter, clay and CEC eff. The data was normalized to sum to 100 %.

I believe the biggest issue the authors must effectively address during revision is the choice of CEC-eff as their explanatory variable of choice. SOM often accounts for a very large portion of the overall CEC-eff of a particular soil sample. Therefore, CEC-eff is dependent on SOM content, not the other way around as the model in the paper suggests (using CEC-eff as an explanatory variable in a model for SOC). That isn't to say that CEC-eff couldn't be couldn't be used as a proxy for SOC content, but it would seem more effective to just measure SOC content since CEC-eff only exhibits a moderate correlation with SOC and is just as laborious to measure. The dependence of CEC on SOM content would explain why correlations among CEC-eff and SOC are stronger in surface soils where SOC is more abundant, as is stated in the discussion. To some extent, the same argument could be made against the findings of Rasmussen et al., 2018, since exchangeable Ca comes from organic exchange sites not associated with mineral surfaces as well as from organo-mineral cation bridging. I believe the use of exchangeable Ca is somewhat more defensible since its role in SOM stabilization is understood on a mechanistic level. It forms cation bridges between organic and inorganic surfaces through ligand exchange. Monovalent cations do not form cation bridges, and therefore cannot contribute to the stabilization of SOM. Exchangeable Mg does not lend itself to stable cation bridges due to a smaller ionic radius. The authors will have to justify from a mechanistic perspective, how CECeff functions to promote SOM accumulation and/or stability. Here I can say that I tested how much of the CEC is affected by Clay and SOM. The introductory material suggests that CEC-eff might act as an effective integrative proxy for properties such as surface area, short-range-order mineral content, clay con-tent and soil organic matter. Here again, is a circular argument. The authors are stating that variance in reactive mineral surface area and SOM exchange sites can be predicted by changes in CEC. They then claim that changes in SOC can be predicted by changes in CEC. We all know that SOM and SOC are inherently linked, and CEC is highly dependent on SOM, so why bother with the proxy? Just measure SOM, which will basically give you a SOC value. Also, explanatory variables included in soil C models must have predictive capacity in order to be useful. We need explanatory variables that will be able to predict how SOC stocks will change in abundance or stability. Because CEC is so heavily influenced by SOM concentration, it changes as a result of changes in SOM concentration, not the other way around. Yes, they are correlated to some degree, but I believe CEC is the dependent variable and SOM (and therefore also SOC) is the explanatory variable. Also, if the desire is to prove that CEC can be used as an integrative proxy for stabilization mechanisms, then the wrong model has been constructed. In order to prove that CEC is accounting for variation in oxalate-extractable metals, clay content, and sur-face area, a model would have to be constructed with CEC as the dependent variable, and oxalate-extractable metals, clay, surface area, etc. as the explanatory variables. It seems like the first hypothesis of this paper should be, "CEC-eff serves as an effective integrative proxy for variables such as metals, clays, and surface area". The authors would then prove the correctness of that hypothesis by using a statistical model to link variation in CEC with variation in metals, clays, and surface area. Then the argument would follow that CEC is much easier to measure than these other properties, as stated in the introduction, and the second hypothesis would then

follow, "Because CEC is an effective integrative proxy of SOM stabilization mechanisms, CEC can be used to predict changes in the stability and abundance of SOC". Then a model similar to the one currently presented would be appropriate.

Our response to the previous comment provides a mechanistic explanation reasoning the hypothesized relationship between CEC eff. and SOC, and how we propose to assess the contribution of SOM to CEC eff. in Swiss forest soils. We hope that the referee finds our explanation and analysis suitable to address his/her concerns. To prove that the variation in CEC eff. is significantly related to oxalate-extractable metals and clay content as well as the interaction of these variables with pH and depth in Swiss forest soils we propose to construct linear mixed effect models. We specifically plan to use CEC eff. as dependent variable and all other explanatory variables as fixed effects (depth, SOM (=SOC here), Clay content, oxalate extractable aluminum and iron oxides, pH). Soil profile ID will be used as a random effect. Linear mixed effect models are proposed for this statistical approach to address the non-independent nature of multiple horizons within one soil profile as in Rasmussen et al. (2018). This analysis will be done on a dataset which consists of a subset of soil profiles where iron and aluminum oxides where measured by oxalate extraction in each of the horizons within a soil profile. Preliminary results of this analysis are shown below (Tables a,b,c) and indicate that indeed oxalate extractable aluminum oxides and clay content (and to a smaller extent iron oxides) contribute to the variation of CEC eff. in Swiss forest soils. The results further corroborate the outcome of the partial regression analysis showing that SOM significantly affects the CEC eff. in topsoils (Table b) but not in subsoils (Table c). To note is also that the interaction between SOM and depth was significant for the model run across the complete vertical soil profiles (Table a), and the interaction between SOM and pH was observed not to be significant in the results of the model run for subsoils (Table c).

a) Complete soil profile: Linear mixed effect model testing the effect of soil depth, SOM (here = SOC), Clay, pH, oxalate extractable aluminum and iron oxides, as well as two-way interactions between soil physicochemical properties with depth on CEC eff.

| a)  Term | Degrees of freedom | Sum of squares % | F value | Prob < F |
|---|---|---|---|---|
| **Depth** | 1 | 115.1 | 424.514 | <0.0001 |
| **SOM** | 1 | 32.2 | 118.953 | <0.0001 |
| **Clay** | 1 | 341.9 | 1261.513 | <0.0001 |
| **$Fe_0$** | 1 | 18.6 | 68.635 | <0.0001 |
| **$Al_0$** | 1 | 93.5 | 345.082 | <0.0001 |
| **pH** | 1 | 83.4 | 307.532 | <0.0001 |
| **SOM * Depth** | 1 | 2.1 | 7.666 | <0.01 |
| **Clay * Depth** | 1 | 5.1 | 18.841 | <0.0001 |
| **$Fe_0$ * Depth** | 1 | 0.1 | 0.340 | n.s. |
| **$Al_0$ * Depth** | 1 | 3.7 | 13.819 | <0.0001 |
| **pH * Depth** | 1 | 2.0 | 7.449 | <0.01 |
| **Residuals** | 923 | 250.2 | | |

b)  Topsoil 0-30 cm depth: Linear mixed effect model testing the effect of SOM (here = SOC), Clay, pH, oxalate extractable aluminum and iron oxides, as well as two-way interactions between soil physicochemical properties with pH on CEC eff.

| b) Term | Degrees of freedom | Sum of squares % | F value | Prob < F |
|---|---|---|---|---|
| pH | 1 | 157.37 | 626.613 | <0.0001 |
| SOM | 1 | 102.07 | 406.437 | <0.0001 |
| Clay | 1 | 87.28 | 347.533 | <0.0001 |
| $Fe_0$ | 1 | 0.01 | 0.059 | n.s. |
| $Al_0$ | 1 | 19.52 | 77.732 | <0.0001 |
| SOM * pH | 1 | 0.32 | 1.274 | n.s. |
| Clay * pH | 1 | 12.47 | 49.658 | <0.0001 |
| $Fe_0$ * pH | 1 | 4.17 | 16.617 | <0.0001 |
| $Al_0$ * pH | 1 | 0.48 | 1.914 | n.s. |
| Residuals | 576 | 144.66 | | |

c) Subsoils <30 cm depth: Linear mixed effect model testing the effect of SOM ( here = SOC), Clay, pH, oxalate extractable aluminium and iron oxides, as well as two-way interactions between soil physicochemical properties with pH on CEC eff.

| c) Term | Degrees of freedom | Sum of squares % | F value | Prob < F |
|---|---|---|---|---|
| pH | 1 | 129.04 | 473.520 | <0.0001 |
| SOM | 1 | 0.80 | 2.922 | n.s. |
| Clay | 1 | 154.05 | 565.298 | <0.0001 |
| $Fe_0$ | 1 | 0.01 | 0.049 | n.s. |
| $Al_0$ | 1 | 26.98 | 99.004 | <0.0001 |
| SOM * pH | 1 | 0.50 | 1.834 | n.s. |
| Clay * pH | 1 | 0.02 | 0.067 | n.s. |
| $Fe_0$ * pH | 1 | 0.86 | 18.841 | n.s. |
| $Al_0$ * pH | 1 | 5.32 | 3.149 | <0.0001 |
| Residuals | 339 | 92.38 | | |

I believe the modeling work in the manuscript could be improved by a slightly different statistical approach. It doesn't seem appropriate to test for the significance of the explanatory variables for the 0-120 cm models without taking depth into account. Depth is a confounding variable due to the fact that most soil physicochemical characteristics vary predictably with depth. The chosen approach then was to split surface soils from subsurface soils (0-30 cm and 30-120 cm) to examine how the relative influence of difference explanatory variables varied with depth. I believe a more appropriate approach would be to apply a linear mixed model using all the explanatory variables as fixed effects. Depth and all its interaction terms would also be included as fixed effects, with SOC as the dependent variable. The resulting model would indicate which of the climatic or physicochemical variables varied in their influence with depth (which fixed effects were significant). The two-way interaction terms that are significant should be fairly easy to interpret given how the data has been transformed. I'm also confused about why pH and its possible interaction terms were not tested for significance. One of the main findings of the paper is that the relative importance of explanatory variables depend on pH. Perhaps pH could be a more useful proxy than one or more of the othe rvariables currently used in the model? Was pH ever included in a model? Are there other soil physicochemical properties that the authors could explore in lieu of CECeff?

This comment was very helpful for the preliminary re-analysis and we adopted the suggested model construction to test the significance of physicochemical variables accounting for the variation in CEC eff.

In our response to the comments of Referee 3 we provide further information on how we propose to modify the discussion and interpretation of our results.

I would also ask the authors to explain their choice of environmental parameters. Why use LAI instead of NPP? Many soil scientists would argue that NPP would be a better predictor of OM inputs to the soil. Why use MAT and MAP instead of PET or a soil moisture regime index? The authors indicate that differences in moisture are important regulators of the downward propagation of C in these soils, because of differences in leaching depth. PET and/or a soil moisture index would do a better job of representing leaching potential because the seasonality and form of precipitation matters, not just the total amount of precipitation.

We appreciate this comments, however the choice of environmental parameters was bound to the variables available for the adopted datasets. We will specify this in the methods section of the paper.

**Anonymous Referee #2

This MS is succinct and well written and uses a robust data set to test their central question.
We thank the Referee for having reviewed our manuscript and for the encouragement in using the selected dataset for analysis.

There appears to be a critical flaw in this analysis – organic matter is likely contributing the majority of CEC in a large number of the samples. CEC at low pH is dominated by permanent charge on clay minerals but as pH rises variable charge on clay and organic matter take over as the dominant control on CEC. I think the results presented in Fig 1 and 3 are a nice demonstration of this phenomena. Given the large range in OM in these samples, it is highly probable that variable charge of OM is driving the correlations seen in this analysis. Unless there is a way of removing the confounding influence of OM on CEC especially at more neutral to basic pH levels, I do not see how this study is publishable. Alternatively, the authors can provide a very convincing argument as to why the bio-geochemistry community should accept the findings. Perhaps I'm just dense and don't get it, but, if that is the case, then the authors need to spend some time in the introduction laying out the logic behind how CECeff is not confounded by OM content in the context of this analysis.

We can understand that the referee was hesitant about the fact that the variable charge of soil organic matter might be leading most of the correlations seen in this analysis. We intend to thoroughly address this issue and provide a mechanistic reasoning (in the introduction section) of why it could make sense to test whether the effective cation exchange capacity (CEC eff.) may be used as a predictor of the potential soil organic carbon (SOC) preservation in Swiss forests. This is certainly only reasonable for soils profiles or depth increments where soil organic matter (SOM) contributes little or nothing to the overall CEC eff. For instance, the contribution of soil organic matter to CEC has been shown by Parfitt et al. (1995) to be greater in surface horizons than at deeper soil depth for several different soil

orders at pH 7. Moreover, Helling et al. (1964) clearly indicated that the contribution of the charge of soil organic matter (SOM) depends on the pH of the soil, with a greater mean relative contribution of organic matter to CEC increasing from pH 2.5 to pH 8. Considering these observations it seems plausible to hypothesize that CEC eff. might be a possible predictor of SOC preservation at specific soil increments falling in specific ranges of pH values in which SOM contributes little or nothing to the CEC eff, such as subsoils. A mechanistic explanation of how we think CEC eff. may influence SOC preservation is provided in our response to Referee 1 (please see our response to previous comments above).

To unravel the confounding effect of organic surfaces on CEC eff. in Swiss forest soils we propose to use partial regression coefficient values combined to the content of soil organic matter, clay and CEC eff. The preliminary results are presented in Figure 1 (above, in the response to Referee 1) and show that in topsoils SOM contributes between 35% and 50% to CEC eff. of the soil and hence that CEC eff. cannot not be used as a predictor of SOC preservation for surface soil horizons. However, the contribution of SOM to CEC eff. in subsoils was lower, ranging between 0 and 11% in soils with pH lower or equal 6.5 and between 17 and 34% for soils with higher pH. Indicating that it is reasonable to test whether CEC eff. could be used as a predictor of potential SOC preservation in subsoils of Swiss forests, particularly those soils with a pH lower or equal to 6.5. We further propose to construct linear mixed effect models to assess how a range of soil variables influence the variation of CEC eff. (soil depth, clay content, iron and aluminum oxides, SOM, pH). The preliminary results are shown in Tables a,b,c (above, in the response to Referee 1) and nicely corroborate the outcome of the partial regression analysis showing that SOM significantly affects the CEC eff. in topsoils (Table b) but not in subsoils (Table c). To note is also that the interaction between SOM and depth was significant for the model run across the complete vertical soil profiles (Table a), and the interaction between SOM and pH was observed not to be significant in the results of the model run subsoils (Table c).

We hope that the referee finds our mechanistic reasoning and re-analysis suitable to address his/her concerns about this study. In our response to Referee 3 we provide further information on how we propose to modify the discussion and interpretation of our results.

A few other comments:

L40 Does correlation with a physiochemical property mean more potential for preservation?

Thank you for this comment, this point will be changed throughout the text specifying that we test whether CEC eff. could be used as a predictor for SOC, but only in those soils profiles or depth increments where SOM contributes little or nothing to the overall CEC eff, such as subsoils.

L113-119 Measurement of Al and Fe forms is equally laborious as calculating CEC (both involve similar extraction protocols then quantification on an ICP or AAS or similar), so this argument is a bit of a red herring

This is true. We will change this argument while rewriting the text.

L160 Exchangeable cations measured from an unbuffered solution will overestimate Ca in calcareous soils and overestimate Na in sodic soils (probably aren't any in Swiss forests), therefore CEC calculated by summing cations instead of by further displacement of the NH4 is not reliable for these soil types.

Yes, it is possible that the Ca contents in calcareous soil layers might be overestimated. This will nevertheless not affect the main result of the study. Moreover, the extraction of exchangeable cations by NH4Cl is a standard method.

L238 Is mmolc/kg an acceptable unit? I thought ccmolc/kg was the standard

The unit mmolc/kg is an acceptable unit and often used in the literature (e.g. Droge and Goss, 2013; Walthert et al., 2013).

**Anonymous Referee #3

Overall, this is an important analysis of proxies that can predict SOC content. This manuscript fits well with recent syntheses and reviews like Rasmussen et al. 2018 and Rowley et al. 2018 (both in Biogeochemistry). The authors tested the ability of CEC, clay, LAI, MAT, and MAP to predict the weighted average SOC content in the surface, subsurface, and whole soil profile in >1000 forest soil profiles across Switzerland. They found that effective CEC was the best predictor of SOC content at higher pH's in the whole soil profile and surface 30 cm of the soil profile while MAP was a stronger predictor at lower pH in the whole profile and surface soil. For the subsoil, both climate variables (MAT and MAP) were the strongest predictors likely due to greater weathering and leaching of organic molecules through the soil profile.

The statistics are sound and clearly presented. The figures are clear, though figures 2and 3 are a bit redundant in that they show the same trends. I think Figure 3 is clearer and suggest only using that figure, but given this is an online open access journal, I see no harm in including both if the authors feel strongly about including both

We thank the referee for the careful analysis of the paper and the constructive comments. It is true that those two figures show very similar results. Since after revising the manuscript we propose to add some other figures it is a good idea to only keep Figure 3 and not show Figure 2 of the manuscript under discussion. Moreover, we suggest to only show 'panel c' of Figure 3 since it appears that it only makes sense to test the use of CEC eff. as a predictor for the potential preservation of soil organic carbon (SOC) in subsoils but not in topsoils (please see our responses above to the comments of Referees 1 & 2).

My largest comment is that I wonder how single cations would be as predictors, suchas Ca and Fe or Al as in Rasmussen et al. 2018 as it would be interesting to test those findings with a different dataset. For example, I wonder how much the strong CEC relationship at higher pH is driven by the Ca cations alone and whether Al ions would better explain the MAP relationship at low pH. To test individual cations would likely illuminate mechanisms better and they may even have stronger relationships with SOC than CEC, but individual cations would then not be the integrative proxy that the authors are seeking.

We thank the referee for this very helpful suggestion. We have now made a new Figure (see Figure 2 below) illustrating the relative contribution of exchangeable Ca2+ and Al3+ as well as other cations to the effective cation exchange capacity (CEC eff.) across the different pH classes for subsoils. In subsoils, 60 to 82 % of the CEC eff. was reflected by exchangeable Ca2+ cations at pH levels higher or equal to 5.5. Instead, in acidic soils Al3+ contributed to about half of the CEC eff. Exchangeable Fe was only poorly correlated to CEC eff. (pearson correlation, r=0.152) and contributed to less than 1% of the variability in CEC eff. across e pH classes.

[Figure]

Figure 2: Relative contribution of Exchangeable Ca, Al and other cations (Na+, K+, Mg2+, Mn2+, Fe2+, H+ ) to CEC eff. in subsoils (30-120 cm depth) of Swiss forests.

Please note that we could also add the contribution of the exchangeable $Na^+$, $K^+$, $Mg^{2+}$, $Mn^{2+}$, $Fe^{2+}$, $H^+$ to this figure if the Editor or Referees find it appropriate. We are cautious as it might overload the figure.

We have also reworked Figure S1 which now also shows the distribution of exchangeable Ca in addition to that of CEC eff., exchangeable Al and SOC in topsoils (0-30 cm depth) and subsoils (30-120 cm depth). From this figure it is clear that the CEC eff largely follows the distribution of exchangeable Ca in alkaline soils.

[Figure]

[Figure]

[Figure]

Figure S1: Median values of effective cation exchange capacity (brown squares), exchangeable Ca (green diamonds), exchangeable Al (orange traingles) and SOC content (grey circles), in topsoils (0-30 cm depth) and subsoils (30 -120 cm) of Swiss forests.

Our initial analysis revealed that the relative control of climatic, vegetation and edaphic variables on SOC evolves as a function of soil pH,  with subsoil SOC content explained mainly by climate at low pH and CEC eff. explaining additional variability (> 35 %) at higher pH values (pH >6.5 in) (Figure 3 below).

[Figure]

Figure 3: Relative contribution of climatic, vegetation and edaphic variables in predicting SOC across pH classes in subsoils (30-120 cm depth). The response variance explained by each environmental variable was normalized to sum to 100 %. MAP, mean annual precipitation (mm), MAT, mean annual temperature (°C), LAI, leaf area index, clay, percentage of clay (%), CEC, effective cation exchange capacity (mmolc kg$^{-1}$).

→ CONCLUSIONS: In topsoils SOM contributes between 35% and 50% to CEC eff. of the soil and hence CEC eff. cannot not be used as a predictor of SOC preservation for surface soil horizons. The contribution of SOM to CEC eff. in subsoils was lower, ranging between 0 and 11% in soils with pH lower or equal 6.5 and between 17 and 34% for soils with higher pH. Indicating that CEC eff. could be potentially used as a predictor of SOC preservation in subsoils of Swiss forests, particularly those soils with a pH lower or equal to 6.5. Considering that CEC eff. has the highest explanatory power in soils with pH higher than 6.5, and that most of the variability in CEC eff. is explained by exchangeable Ca in soils with these pH we suggest that exchangeable Ca (for which SOM stabilization is better understood on a mechanistic level than CEC.eff.) may be better used to describe the variation in SOC in alkaline forest soils

Our results further suggest that the strong influence of MAP and MAT on SOC is likely partially related to the interaction of climate with soil chemical properties. For instance, in acid soils the presence of Al$^{3+}$ cations is commonly associated to the production of organo-metal complexes during the process of weathering. Secondly, a strong chelation of Al$^{3+}$ by organic acids in Swiss forest soils characterized by a high MAP and acid soils, is pointed out by the high levels of exchangeable aluminium (Al$^{3+}$) found at low pH classes (Fig. S1), and by the observation of the highest MAP regimes in regions with the most acidic soils (Fig. S2, S3 presented in the paper under discussion). These patterns corroborate previous results showing that the relatively humid climate of the Southern Alps promotes both the formation of pedogenic oxides and the

leaching of dissolved organic C from the organic layer to the mineral soil, where organic matter i.e. tannins and other polyphenols forms organo-metal complexes (Eckmeier et al., 2010). Moreover, in soils with low pH and high levels of water availability, iron oxides are dissolved and Fe becomes exchangeable (Schwertmann, 1991) and can interact with SOC protecting it. All in all, these observations are in agreement with the recent syntheses and reviews by Rasmussen et al. (2018) and Rowley et al. (2018), which highlighted the importance of considering pH in determining physicochemical drivers of SOC (i.e. Ca, Al or Fe).

Lastly, I am not sure if it belongs in the Introduction or Discussion, but Rowley's 2018synthesis, "Ca-mediated stabilization of organic carbon" should be cited in this paper as it also touches upon pH differences in the controls on SOC content. Figure 3 is particularly relevant.

Thank you for this comment. We are planning to cite Rowley's 2018 synthesis paper throughout the introduction and discussion of our revised manuscript.

I have some minor comments where the manuscript needs some clarification and where the findings of Rasmussen et al. could be more accurately presented. Specific comments:

Abstract L23: delete "as compared to the mere quantification of clay-sized particles "because as you stated in the intro, that is not a trivial analysis to do.
Introduction:L79: Please clarify "exchangeable Ca and Fe, and Al oxyhydroxides". Rasmussen et al. tested the predictive capabilities of oxalate extractable Fe and oxalate extractable Al. These are measures of organo-metal complexes and short range order minerals, not exchangeable Fe or all Al oxyhydroxides. L83: Add that soil pH also determinest he relative charges of organic molecules and soil minerals and thus the likelihood that organic molecules will sorb to minerals, so not just organo-metal complexes with ligands. L84: The line about depth should be a separate sentence as it is unclear herewhat you want to emphasize about soil depth.

We will appropriately revise the manuscript and correct these points as suggested.

L114-116: I disagree that Fe and Al variables cannot be measured on large datasets. Doesn't the analysis in Rasmussen et al. of a large soil dataset, which you cite for this sentence, contradict that statement?

This is true. We will remove this argumentation. CEC. eff is nevertheless routinely measured for agricultural and forestry monitoring purposes.

Methods: L147: please define "fine earth" does this mean, you corrected for rocks? How did you classify "fine"?

Fine earth: all particles smaller than 2 mm, the threshold of 2 mm is a pedological standard many chemical analyses are made by use of the fine earth fraction

(moreover, the density of the fine earth is the density of all particles smaller than 2 mm)

L185: What time period during the growing season was used to determine LAI?

The Leaf Area Index (LAI) was calculated according to Schleppi et al. 2011 based on cover abundance data from vegetation surveys. All surveys were made during the vegetation period between June and September. We will provide this information in the manuscript. It would also be possible to provide exact dates of the surveys.

L218: The line "for each statistical test P<0.05 was . . ." directly contradicts the previous statement. Maybe write, "For all other statistical tests . . ."

This sentence will be changed accordingly.

Results: When reading the results, my first question was what the distribution of samples among the different pH classes were. You might want to move up Figure 4 to the beginning as it strengthens the interpretation of your analyses to know how evenly distributed the samples were among pH classes.

As suggested, we will first show Figure 4 before Figure 3 of the manuscript under discussion.

Discussion: Line 299: Here is a good place to cite Rowley et al. 2018. L314-317:This statement here leads to a bit of a 'chicken and egg' conundrum. Is there more Ca because negative charges on OM can bind to it or is there more SOM because there is more Ca to bind to it? Maybe it doesn't matter for predictions. I have the same issue for when pyrophosphate extractable Fe and Al are used to predict SOC as that extraction targets organo-mineral complexes. Maybe the Ca stabilization mechanisms brought up by Rowley could help here.

*** Text below copied from a correction that Referee 3 made on 14 April 2019 to his comment ***

Just realized I was really thinking too much about Ca in my comment! By the chicken and egg conundrum I meant to say CEC, which is as reviewer 2 pointed out, confounded by SOM. Unlike reviewer 2, I don't think this makes it unpublishable but it maybe worth looking at Ca only as a proxy as well to help sort it out!

We can understand that the referee was hesitant about the fact that the variable charge of soil organic matter might be confounding the correlations seen in this analysis. We intend to thoroughly address this issue and provide a mechanistic reasoning (in the introduction section) of why it could make sense to test whether the effective cation exchange capacity (CEC eff.) may be used as a predictor of the potential preservation of SOC in Swiss forests. This is certainly only reasonable for soils profiles or depth increments where soil organic matter contributes little or nothing to the overall CEC eff, such as subsoils. In our responses to Referees 1 and 2 we provide 1) a mechanistic explanation reasoning the hypothesized relationship, and 2) information on how we can address the confounding effect of organic surfaces (please see our responses above to the comments of Referees 1 and 2).

L317: Tone down this statement to "may be instead" in place of "is instead" as to know for sure you would need a mechanistic test as you nicely point out below. L367-360: Please reword this sentence. I found this sentence to be confusing in its structure, particularly the clause in dashes, and had to reread it several times.

These changes will be made in the revised version.

Cited references:

Chen, S., Arrouays, D., Angers, D. A., Chenu, C., Barré, P., Martin, M. P., Saby, N. P. A., and Walter, C.: National estimation of soil organic carbon storage potential for arable soils: A data-driven approach coupled with carbon-landscape zones, Science of The Total Environment, 666, 355-367, https://doi.org/10.1016/j.scitotenv.2019.02.249, 2019.

Doetterl, S., Stevens, A., Six, J., Merckx, R., Van Oost, K., Casanova Pinto, M., Casanova-Katny, A., Muñoz, C., Boudin, M., Zagal Venegas, E., and Boeckx, P.: Soil carbon storage controlled by interactions between geochemistry and climate, Nature Geoscience, 8, 780, 10.1038/ngeo2516

https://www.nature.com/articles/ngeo2516#supplementary-information, 2015.

Droge, S. T. J., and Goss, K.-U.: Sorption of Organic Cations to Phyllosilicate Clay Minerals: CEC-Normalization, Salt Dependency, and the Role of Electrostatic and Hydrophobic Effects, Environmental Science & Technology, 47, 14224-14232, 10.1021/es403187w, 2013.

Eckmeier, E., Egli, M., Schmidt, M. W. I., Schlumpf, N., Nötzli, M., Minikus-Stary, N., and Hagedorn, F.: Preservation of fire-derived carbon compounds and sorptive stabilisation promote the accumulation of organic matter in black soils of the Southern Alps, Geoderma, 159, 147-155, https://doi.org/10.1016/j.geoderma.2010.07.006, 2010.

Helling, C. S., Chesters, G., and Corey, R.: Contribution of organic matter and clay to soil cation-exchange capacity as affected by the pH of the saturating solution, Soil Science Society of America Journal, 28, 517-520, 1964.

Klamt, E., and Sombroek, W.: Contribution of organic matter to exchange properties of Oxisols, 1988.

Luo, Z., Feng, W., Luo, Y., Baldock, J., and Wang, E.: Soil organic carbon dynamics jointly controlled by climate, carbon inputs, soil properties and soil carbon fractions, Global Change Biology, 23, 4430-4439, 10.1111/gcb.13767, 2017.

Luo, Z., Wang, G., and Wang, E.: Global subsoil organic carbon turnover times dominantly controlled by soil properties rather than climate, Nature Communications, 10, 3688, 10.1038/s41467-019-11597-9, 2019.

Minick, K. J., Fisk, M. C., and Groffman, P. M.: Soil Ca alters processes contributing to C and N retention in the Oa/A horizon of a northern hardwood forest, Biogeochemistry, 132, 343-357, 2017.

Parfitt, R. L., Giltrap, D. J., and Whitton, J. S.: Contribution of organic matter and clay minerals to the cation exchange capacity of soils, Communications in Soil Science and Plant Analysis, 26, 1343-1355, 10.1080/00103629509369376, 1995.

Rasmussen, C., Heckman, K., Wieder, W. R., Keiluweit, M., Lawrence, C. R., Berhe, A. A., Blankinship, J. C., Crow, S. E., Druhan, J. L., Hicks Pries, C. E., Marin-Spiotta, E., Plante, A. F., Schädel, C., Schimel, J. P., Sierra, C. A., Thompson, A., and Wagai, R.: Beyond clay: towards an improved set of variables for

predicting soil organic matter content, Biogeochemistry, 137, 297-306, 10.1007/s10533-018-0424-3, 2018.

Rowley, M. C., Grand, S., and Verrecchia, É. P.: Calcium-mediated stabilisation of soil organic carbon, Biogeochemistry, 137, 27-49, 2018.

Schwertmann, U.: Solubility and dissolution of iron oxides, Plant and Soil, 130, 1-25, 10.1007/bf00011851, 1991.

Sposito, G.: The chemistry of soils, Oxford university press, 2008.

Sutton, R., Sposito, G., Diallo, M. S., and Schulten, H. R.: Molecular simulation of a model of dissolved organic matter, Environmental Toxicology and Chemistry: An International Journal, 24, 1902-1911, 2005.

Vos, C., Don, A., Hobley, E. U., Prietz, R., Heidkamp, A., and Freibauer, A.: Factors controlling the variation in organic carbon stocks in agricultural soils of Germany, European Journal of Soil Science, 70, 550-564, 10.1111/ejss.12787, 2019.

Walthert, L., Graf Pannatier, E., and Meier, E. S.: Shortage of nutrients and excess of toxic elements in soils limit the distribution of soil-sensitive tree species in temperate forests, Forest Ecology and Management, 297, 94-107, https://doi.org/10.1016/j.foreco.2013.02.008, 2013.